# Blood pH Analysis in Combination with Molecular Medical Tools in Relation to COVID-19 Symptoms [note 1]

**DOI:** 10.3390/biomedicines11051421

**Published:** 2023-05-11

**Authors:** Hans-Christian Siebert, Thomas Eckert, Anirban Bhunia, Nele Klatte, Marzieh Mohri, Simone Siebert, Anna Kozarova, John W. Hudson, Ruiyan Zhang, Ning Zhang, Lan Li, Konstantinos Gousias, Dimitrios Kanakis, Mingdi Yan, Jesús Jiménez-Barbero, Tibor Kožár, Nikolay E. Nifantiev, Christian Vollmer, Timo Brandenburger, Detlef Kindgen-Milles, Thomas Haak, Athanasios K. Petridis

**Affiliations:** 1RI-B-NT—Research Institute of Bioinformatics and Nanotechnology, Schauenburgerstr. 116, 24118 Kiel, Germany; 2Department of Chemistry and Biology, University of Applied Sciences Fresenius, Limburger Str. 2, 65510 Idstein, Germany; 3RISCC—Research Institute for Scientific Computing and Consulting, Ludwig-Schunk-Str. 15, 35452 Heuchelheim, Germany; 4Institut für Veterinärphysiologie und Biochemie, Fachbereich Veterinärmedizin, Justus-Liebig Universität Gießen, Frankfurter Str. 100, 35392 Gießen, Germany; 5Department of Biophysics, Bose Institute, P-1/12 CIT Scheme VII (M), Kolkata 700054, India; 6Department of Biomedical Sciences, University of Windsor, Windsor, ON N9B 3P4, Canada; 7Institute of BioPharmaceutical Research, Liaocheng University, Liaocheng 252059, China; 8Klinik für Neurochirurgie, Alfried Krupp Krankenhaus, Rüttenscheid, Alfried-Krupp-Straße 21, 45131 Essen, Germany; 9Klinik für Neurochirurgie, Klinikum Lünen, St.-Marien-Hospital, Akad. Lehrkrankenhaus der Westfälische Wilhelms—Universität Münster, 44534 Lünen, Germany; 10Institute of Pathology, University of Nicosia Medical School, 2408 Egkomi, Cyprus; 11Department of Chemistry, University of Massachusetts Lowell, 1 University Avenue, Lowell, MA 01854, USA; 12CIC bioGUNE, Bizkaia Technology Park, Building 800, 48160 Derio, Spain; 13Center for Interdisciplinary Biosciences, Technology and Innovation Park, P. J. Šafárik University, Jesenná 5, 04001 Košice, Slovakia; tibor.kozar@upjs.sk; 14Laboratory of Glycoconjugate Chemistry, N. D. Zelinsky Institute of Organic Chemistry, Russian Academy of Sciences, Leninsky Prospect 47, 119991 Moscow, Russia; 15Department of Anesthesiology, University Hospital Düsseldorf, Heinrich-Heine University Duesseldorf, Moorenstr. 5, 40225 Düsseldorf, Germany; 16Diabetes Klinik Bad Mergentheim, Theodor-Klotzbücher-Str. 12, 97980 Bad Mergentheim, Germany; 17Medical School, Heinrich-Heine-Universität Düsseldorf, Universitätsstr. 1, 40225 Düsseldorf, Germany; opticdisc@aol.com

**Keywords:** SARS-CoV-2, coronavirus fusion peptides, incretin mimetics, blood pH, molecular modeling, NMR

## Abstract

The global outbreak of SARS-CoV-2/COVID-19 provided the stage to accumulate an enormous biomedical data set and an opportunity as well as a challenge to test new concepts and strategies to combat the pandemic. New research and molecular medical protocols may be deployed in different scientific fields, e.g., glycobiology, nanopharmacology, or nanomedicine. We correlated clinical biomedical data derived from patients in intensive care units with structural biology and biophysical data from NMR and/or CAMM (computer-aided molecular modeling). Consequently, new diagnostic and therapeutic approaches against SARS-CoV-2 were evaluated. Specifically, we tested the suitability of incretin mimetics with one or two pH-sensitive amino acid residues as potential drugs to prevent or cure long-COVID symptoms. Blood pH values in correlation with temperature alterations in patient bodies were of clinical importance. The effects of biophysical parameters such as temperature and pH value variation in relation to physical-chemical membrane properties (e.g., glycosylation state, affinity of certain amino acid sequences to sialic acids as well as other carbohydrate residues and lipid structures) provided helpful hints in identifying a potential Achilles heel against long COVID. In silico CAMM methods and in vitro NMR experiments (including ^31^P NMR measurements) were applied to analyze the structural behavior of incretin mimetics and SARS-CoV fusion peptides interacting with dodecylphosphocholine (DPC) micelles. These supramolecular complexes were analyzed under physiological conditions by ^1^H and ^31^P NMR techniques. We were able to observe characteristic interaction states of incretin mimetics, SARS-CoV fusion peptides and DPC membranes. Novel interaction profiles (indicated, e.g., by ^31^P NMR signal splitting) were detected. Furthermore, we evaluated GM1 gangliosides and sialic acid-coated silica nanoparticles in complex with DPC micelles in order to create a simple virus host cell membrane model. This is a first step in exploring the structure–function relationship between the SARS-CoV-2 spike protein and incretin mimetics with conserved pH-sensitive histidine residues in their carbohydrate recognition domains as found in galectins. The applied methods were effective in identifying peptide sequences as well as certain carbohydrate moieties with the potential to protect the blood–brain barrier (BBB). These clinically relevant observations on low blood pH values in fatal COVID-19 cases open routes for new therapeutic approaches, especially against long-COVID symptoms.

## 1. Introduction

Molecular medicine entails the development of new ways to diagnose and treat diseases. The diagnostic and therapeutic approaches of SARS-CoV-2 are outstanding examples of the need to understand the mechanisms and possible treatment procedures on a molecular level. NMR experiments on dodecylphosphocholine (DPC) micelles mixed with gangliosides (e.g., GM3) can be carried out under defined pH and temperature conditions [1]. Such a methodology is also suitable to test and refine potential SARS-CoV-2 inhibitory fusion peptides [2,3,4,5,6] as well as other antimicrobial peptides and proteins such as lysozymes [7,8], hevein domains [9] or defensins [10,11,12] in respect to their membrane affinities and lectin-like functions. Thereby, it is possible to determine the effectiveness of these biomedical molecular structures in the context of clinical data [13,14,15,16]. The micelles mixed with surface-exposed sialic acid containing glycolipids (e.g., GM3, GM1, and GD1A) can be regarded as a simple cell membrane model with sialic acids acting as key contact structures of pathogens [14,17] as well as contact structures on host cell surfaces [18,19,20,21,22,23,24,25]. NMR studies supported by molecular modeling calculations enable a detailed analysis of the interactions between proteins or peptides with other biomolecules [26,27,28,29,30], lipids [8,26,31,32] oligosaccharides or glycoconjugates [18,19,20,33] under various physiological conditions. This strategy was previously successfully applied when collagen fragments and sulfated glucosamines were tested and compared in relation to their receptors [34,35,36,37,38].

In the case of the SARS-CoV-2 spike protein—ACE2 receptor complex—a potential pH-dependent influence of Zn^2+^ cations which stabilizes the bio-active conformation of this complex is of particular interest [39]. Computational studies clearly show that a Zn^2+^ cation contributes in a pH-dependent manner to the structural dynamics of the surrounding His and Glu residues in the virus spike protein–ACE2 receptor complex. Interactions of the ACE2 receptor which occur in cells present in lung tissue can be significantly influenced by crucial pH values of approximately pH 6. These values correspond to the physiological conditions on lung tissue surfaces [40,41]. However, other organs whose tissue pH values show a higher pH value than the lung can also be affected by pH value alterations in the range of the blood pH which is typically between pH 7.35–7.45. In this pH range, His residues are still partly protonated. In this context, the stability of the blood–brain barrier (BBB) and its collagen network is of highest importance [15,16]. Therefore, we evaluated blood pH values from patients at an intense care unit. Blood pH values below pH 7.35 correspond to acidosis while blood pH values above pH 7.45 indicate alkalosis. A clinically relevant question arose on whether the blood pH value alterations of patients treated in intense care units can be correlated with the severeness of this disease and the fate to become a survivor or an exitus (fatal case). Furthermore, these observations of clinical data are regarded as a starting point for in vitro NMR experiments with model membranes mixed with the ganglioside GM1 and potential antiviral peptides [9,25,42,43,44,45,46,47,48,49]. Thus, two physical parameters that were found to be characteristic of patients with different outcomes, blood pH and body temperature, were altered in the in silico molecular modeling studies and during the in vitro NMR experiments in which the solution properties and oligosaccharide affinities of incretin mimetics with one or two His residues were tested. Dodecylphosphocholine (DPC) micelles mixed with gangliosides [1,50] are suited role models because the lectin-like SARS-CoV-2 spike glycoprotein has a specific sialic acid affinity [43,44]. The results of the experiments described here will contribute to the development of improved diagnostics and therapeutic strategies in respect to a new generation of incretin-like inhibitory viral fusion peptides [29,51,52].

The incretins, glucagon-like peptide-1 (GLP-1) and glucose-dependent insulinotropic polypeptide (GIP), are glucose-lowering, intestinal peptide hormones. Their receptors, GLP-1R and GIPR, are primarily localized to pancreatic islet cells and implicated in the treatment of type 2 diabetes mellitus (T2DM). It is important to emphasize that both receptors are also present elsewhere and particularly throughout the nervous system, on the dendritic branches of neurons as well as on activated microglia and astrocytic cells [53,54]. The N-glycosylated incretin receptors are coupled to the cAMP second messenger pathway whose upregulation is associated with neuroprotection and anti-inflammation [55,56,57]. Therefore, the potential therapeutic benefit of incretin mimetics in brain pathologies, especially in relation to long-COVID symptoms is of growing interest. Our hypothesis assumes that incretin mimetics provide excellent models for the development of new antiviral peptides under consideration of pH and temperature effects. Incretin mimetics are suitable molecules due their structural properties being linear peptides with proper amino acid sequences. In addition, receptor-induced neurotrophic, defensive and anti-inflammatory processes of incretin mimetics, being neuroprotective in respect to long-COVID symptoms, will be discussed.

We tested incretin-like model structures which may block SARS-CoV-2 entry based on their known membrane affinity and potential use in oral delivery to patients. The substances and nanoparticle models under study are summarized and visualized in Figure 1, Figure 2 and Figure 3. The methodology used is consistently described in part 4 of the presented article.

The similarities in the symptoms of diabetes, SARS-CoV-2 and long-COVID symptoms have led to the use of incretin mimetics playing a key role in clinical research projects [58,59,60,61,62]. These studies focus on the nanomedical aspects of the corresponding interactions between peptides/proteins, lipids and oligosaccharides.

In our study, the DPC micelles mixed with GM1 gangliosides (sialic acid containing glycolipids) were considered as model membranes of the host cell surfaces. However, sialic acids are also contact structures of virions and virus models due to their presence at the terminus of the saccharide chains [63,64]. In this context, sialic acid-coated silica nanoparticles are well suited virus models because they allow one to study the impact of this carbohydrate moiety on specific residues within the spike-protein of the SARS-CoV-2 virus.

## 2. Results

### 2.1. Clinical Studies

Blood pH values of 25 SARS-CoV-2 patients in an intense care unit were detected every day (12 fatal cases and 13 survivors). The average blood pH values of 25 patients are shown in Figure 1. In this figure, the pH values were detected at exact times per day or per week of the clinical stay of the patients. The pH values of surviving patients are shown in green; fatal cases in red. In Figure 2, the pH values were detected at exact times per day or per week at the clinical stay of the patients. The blood pH values from these 25 patients were listed during these time intervals as (a) minimum pH values and as (b) maximum pH values. Notably, lower blood pH values measured during all days of the stay in the hospital were directly correlated with the risk of fatality. It must be emphasized that the pH measurements during the first 7 days (with a special focus on days 1 to 3) are the most meaningful since the pH values can strongly be influenced by other effects (multiple organ failure) after this time period and thus are only indirectly correlated with the attack of the virus.

A statistical analysis of the blood pH values from 25 patients in the intense care unit was carried out to determine whether the observed pH alterations within this patient group were of statistical significance.

Our clinical data lend support to a model in which pH may be a valuable prognostic marker for patient outcomes in the first three to seven days of clinical treatment. We therefore investigated this further by incorporating these parameters into molecular modeling calculations and NMR experiments of incretin-like potential inhibitory SARS-CoV-2 fusion peptides. In order to examine the feasibility of this assumption, in silico molecular modeling calculations and in vitro NMR experiments were performed under different pH and temperature conditions. We note that when determining treatment options for COVID-19 patients, it is important to understand the mechanics of the interplay between SARS-CoV-2, potential treatments and cell models under the pH and temperature conditions seen in patients more at risk.

### 2.2. CAMM: ACE2 Receptor and Incretin Mimetics

The conformation of the ACE2 receptor (6M0J.pdb) is strongly pH dependent due to its Zn^2+^ ion interacting with His-374 and His-378 as well as with Glu-375, Glu-402 and Glu-406 (see Appendix A). We therefore concentrated on in silico studies on the zinc cation and neighboring amino acids. One two-fold positively charged zinc ion is surrounded by two histidine (His374, His378) residues and three glutamates (Glu375, Glu402, Glu406) residues. The pH value for stabilizing the histidine (His) side chains has to be above 6. At pH values below 6, the zinc cation separates from the positive charged His residues and thus alters the tertiary structure. It is noted that the amino acid residues mentioned above are located on different strands and are therefore apart from each other in respect to the primary sequence.

Interestingly, it is known that the pH dependence of the His residues also plays a crucial role in the context of linear peptides which can act as peptides inhibiting viral fusion when surrounded by Ca^2+^ cations at various concentrations [65]. Regarding the incretin mimetics under study (in our case, role models for inhibitory fusion peptides), we focused on Semaglutide with a primary His residue and Pt2His with a primary and a tertiary pH-sensitive His residue in the amino acid sequence. The Pt2His structure at various possible binding positions to human serum albumin is shown in Figure 3A,B.

The structural dynamics of the lipid chain conformations of Tirzepatide and Semaglutide are similar to that of Pt2his. However, in the case of Pt2His, the lipid chain is linked to K14 and the chain is attached to K20 in the case of Tirzepatide and Semaglutide. The Semaglutide model structure in Figure 4 A was analyzed in the presence of glycosylated DPC micelles in respect to their structural dynamics along with NMR experiments together with MD simulation methods. The structural dynamics of the identical lipid tails of Semaglutide and Pt2His (Figure 4B,C) are independent from the two different positions in the amino acid part of these incretin mimetics. When inhibitory fusion peptides are used as antiviral therapeutics, it is essential that they are guided through the blood stream by a shuttle system to many target organs. In the examples described here, it is human serum albumin (HSA, Figure 3) to which the lipid parts of the two incretin mimetics under study are attached. The lipid part (Figure 5B,C) is linked to Lys20 (K20) in the case of Semaglutide or at Lys14 (K14) as performed for Pt2His. Since the lipid part has a defined impact on the structural dynamics of Pt2His Figure 5A of Tirzepatide (Figure 5B) as well as on Semaglutide (Figure 5C), it was possible to study the corresponding structural influence by NMR and molecular modeling techniques [66,67,68]. This analysis allowed us to study different incretin mimetics which could be used as potential SARS-CoV-2 therapeutics.

In addition, we carried out MD simulations for the SARS-CoV fusion peptides 2RUM and 2RUO in a water or in a lipid environment. We emphasize here that no His residues occur in the two peptides 2RUM and 2RUO. These fusion peptides seem to only interact with the glycosylated cell membrane while the glycosylated GIP—and GLP-1—receptors are the main targets for Pt2His, Semaglutide and Tirzepatide.

Our experimental setting provides a simple membrane model suited for testing SARS-CoV fusion peptides as well as incretin mimetics interacting in a receptor-independent way with a glycosylated model membrane. The correlation of these findings in the context of issues associated with a damaged blood–brain barrier (BBB) are addressed by a detailed analysis of interactions with collagen structures. Furthermore, sialic acid-coated nanoparticles and certain glycosylation techniques can also be applied to test the membrane properties [69,70].

It is known that a decrease in collagen destabilizes the blood–brain barrier [15,16]. Therefore, the role of collagen-structures was taken into account when ACE2 virus spike protein complexes were analyzed. Furthermore, we considered that the ACE2 receptor can also occur as an unbound enzyme free in the blood stream.

In Figure 6A, the Asn-109 covalently linked surface trisaccharide of the ACE2 receptor is shown with green carbon atoms. The visualization reflects the 4APH.pdb structure. In Figure 6B, the ACE2 receptor is depicted in complex with the SARS-CoV-2 virus spike protein (pdb code 2AJF) in a ribbon and surface representation. The surface of the receptor is colored in grey and the surface of the spike protein is highlighted in yellow (Figure 6B).

The binding sites of the angiotensin receptor (violet surface) are the focus of Figure 7. Left side, (A)—SiteMap-calculated binding sites of the angiotensin receptor (the angiotensin molecular surface is colored violet). Right side; (B)—the angiotensin receptor (grey surface) in complex with virus spike protein (yellow surface). The first four SiteMap-preferred binding sites are shown with red ellipses and yellow/blue highlights. An angiotensin receptor (grey surface) and virus spike protein (yellow surface) complex is shown in Figure 7B. The first four SiteMap-preferred collagen binding sites are highlighted with red ellipses and yellow/blue highlights. Conformational changes that occur in protein-collagen complexes which can be influenced by temperature and/or pH alterations are displayed in Figure 8.

The angiotensin receptor protein is modelled in interaction with collagen strands as seen from molecular dynamics (MD) simulations. The carbohydrates linked to the proteins are also highlighted (green carbons). The conformations shown were extracted from 10 ns MD simulation in a water environment. The collagen model is shown as grey colored ribbon for (A) and semitransparent blue and green colored collagen surface is added for (B). Molecular modeling and NMR methods are thus invaluable tools in analyzing important changes that may occur in the blood–brain barrier (BBB). Blood temperature-dependent destabilization of the BBB under conditions of acidosis [16,71] is a potential explanation for pH-dependent effects in respect to the gravity of acute and/or long-COVID symptoms. Since collagen strands are essential building blocks of the blood–brain barrier, we focused on the ACE2-receptor–spike protein–collagen interactions. This allowed us to examine the way that such complexes can be described in respect to their structural dynamics and whether a temperature- and pH-dependent effect could be expected. It is now of importance to find out in which way the insights obtained by clinical observations and molecular modeling can be verified with the help of NMR experiments. In this context, it is important to mention that collagen hydrolysates, which are mixtures of linear peptides and short collagen fragments, have been successfully studied by various NMR methods [34,35,36,37,38]. Clinically, they are orally administered and serve to act as chondroprotective agents in patient target tissues. The design of the NMR experiments flanked by molecular modeling studies as introduced here provide guidelines for cell-culture experiments under certain pH and temperature conditions [72,73]. This will be outlined in detail in a follow-up study with the focus on tissue studies and cell-culture experiments as well as on a mass spectrometric analysis of blood probes from COVID-19 patients.

### 2.3. NMR: Semaglutide and Pt2His

NMR experiments were carried out under consideration of two physiological key-parameters: temperature and pH value. The incretin mimetics Semaglutide and Pt2His as well as the two SARS-CoV fusion peptides 2RUM and 2RUO show detectable model membrane interactions in different ways. These interactions are of interest when looking for inhibitory viral fusion peptides providing a steric perturbation of the entry platform created by the viral fusion peptides. The other way to hinder the spread of SARS-CoV-2 and block viral fusion is the direct interaction of incretin mimetics with their glycosylated receptors.

The incretin mimetics Semaglutide and Pt2His were the first approach to study the temperature-dependence of peptides in the pH range of interest. We speculate that these two peptide drugs may inhibit SARS-CoV-2 infection and virus spread in the body due to membrane-related effects. The peptide membrane interactions were analyzed in the presence of non-glycosylated and glycosylated model membranes. The glycosylated membranes were created with the help of GM1 gangliosides as contact structures of the host cell membrane. Ganglioside GM1 is a glycolipid with one sialic acid residue in which the N-acetyl-neuraminic acid (a member of the sialic acid family) residue ισ α2–3 linked to a galactose unit. Furthermore, silica nanoparticles at the size of the SARS-CoV-2 virus with a sialic acid-coated surface were used as a suited virus standard model for referencing the intensities of our ^31^P NMR signals for glycosylated DPC membranes at 42 °C and 37 °C.

Four ^31^P signals are shown in the left as well as in the right part of Figure 9. The signals represent the micelles of four different experimental settings at two temperatures at a constant physiological pH value of 7.4: 42 °C (high risk temperature of the patients) and 37 °C (normal temperature of the patients). Black signal: pure DPC micelles. Blue signals: DPC micelles with sialic acid containing compounds: GM1 and sialic acid-coated silico nanoparticles. Green signal: Pt2His in the presence of DPC micelles. Red signal: Semaglutide in the presence of DPC micelles.

At 42 °C, the ^31^P signal of DPC micelles in the presence of Pt2His is high and sharp while it changes at 37 °C to a broad and smaller signal. This result indicates that the interaction of micelles and Pt2His is strongly temperature dependent and thereby influenced by the tumbling rate of the micelles. In contrast to the Pt2His result, the Semaglutide DPC micelle interactions show a minor temperature dependence in the physiological range. The black DPC ^31^P signal represents the pure DPC micelles without any peptide or sialic acid containing compounds. It turned out that this signal can be used as a temperature-independent reference signal. DPC micelles mixed with GM1 gangliosides in the presence of sialic-coated nanoparticles as a basic cell membrane model in addition to pure DPC micelles.

We varied the concentrations of both GM1 gangliosides and sialic acid-coated nanoparticles. However, we found that only one concentration already taken for the peptides under study was of importance. A 40 molar access of DPC to the ganglioside GM1 as well as to the sialic acid-coated nanoparticles leads to proper mixtures, Therefore, the molar ratios of the ganglioside GM1, the sialic acid-coated silico nanoparticles, the incretin mimetics and the SARS-CoV fusion peptides to DPC has to be adjusted to 1:40.

The fact that interactions, especially between aromatic amino acid residues and carbohydrate moieties play a prominent role in complex formation independent of the peptides or proteins studied is documented in the literature [18,30,74]. This is of special interest here, because we focus on the amino acid residues that are acting as specific contact structures in the presence of a DPC environment.

The SARS-CoV fusion peptides 2RUM (MWKTPTLKYFGGFNFSQIL-NH2) and 2RUO (GAALQIPFAMQMAYRF-NH2) have provided essential supporting data for the interpretation of the ^31^P NMR results obtained for Semaglutide and Pt2His. These two linear peptides 2RUM and 2RUO do not have a His residue in their amino acid sequence but other aromatic amino acids, especially the CIDNP sensitive Tyr and Trp residues. Similar peptides with a specific oligosaccharide affinity have already been analyzed with NMR, ESI-MS and CAMM methods [18,30]. In the case of the 19-mer peptide 2RUM, we have observed two different ^31^P signals of the DPC micelles in contrast to the ^31^P NMR results from Semaglutide, 2PtHis and the 16-mer peptide 2RUO. The two ^31^P signals of different size argue in favour of two distinct populations of micelles at different size in the presence of 2RUM. The signal spitting does not depend on the two measurement temperatures. The splitting of the signals is an important result because it proves that only one population of micelles exists in the presence of Semaglutide, Pt2His or 2RUO independent of the physiological temperature.

Notably, the four peptides under study and the sialic acid containing silica nanoparticles interfere in characteristic ways with the non-glycosylated and glycosylated DPC model membranes, Figure 10, Figure 11 and Figure 12 and Appendix A.

## 3. Discussion

Our preliminary clinical study indicated that the body temperatures of COVID-19 patients were always well above normal human body temperatures. This was coupled with a significant lowering of blood pH in affected individuals. The observed pH alterations gave us an important hint that the two His residues of a new incretin prototype at position one and three could influence the incretin properties in a pH-dependent way. This could be tested in a combination of NMR experiments with CAMM (computer-aided molecular modeling). Given this, we incorporated these clinical parameters into our NMR experiments of established and novel incretin mimetics and their lipid tails with a high HSA affinity. Although the number of blood samples was not large, we emphasize that these observations were statistically significant.

HSA (human serum albumin) is a well-known shuttle system for incretin mimetics. In our context, it was essential to use CAMM to analyze the suitability of HSA as a transporter for our new incretin prototype on a sub-molecular level.

Incretins and incretin mimetics bind to incretin receptors in a specific way. The impact of these interactions on gastro-intestinal and neurological processes is well studied (as well as discussed and cited above). The gravity of a SARS-CoV-2 infections is closely related to that of other systemic diseases such as diabetes. Therefore, a positive influence of these medications on the course of a coronavirus infection can in particular be explained by the corresponding receptor interactions [75,76]. Furthermore, linear SARS-CoV-2 peptides seem to stabilize or destabilize the virus host–cell interactions without a specific receptor. They directly trigger the cell membrane in order to support or block the entry of the virus. To understand this process, we used a simple DPC ganglioside membrane model and applied ^31^P NMR experiments for the incretin mimetics and the viral fusion peptides under study. It turned out that the signal shifts and line broadening regarding the corresponding ^31^P NMR signals of DPC clearly indicate significant differences between the interactions of the peptides under study and the glyco-membrane model.

A contributing element to fatal SARS-CoV-2 cases may be pH- and temperature-dependent destabilization of the blood–brain barrier (BBB) under conditions of acidosis [16,71]. An effect on the brain vasculature is suspected since neurological symptoms and vascular lesions are seen in COVID-19 infections. It is important to note that that the ACE2 receptor is ubiquitously expressed in vessels of the cerebral cortex. Furthermore, the spike protein ACE2 receptor interaction is known to be correlated with property changes within the BBB [77].

The infection with SARS-CoV-2 in relation to the ACE2 receptor is not limited to affecting the blood–brain barrier (BBB), it also affects ACE2 receptor rich cells of the choroid plexus epithelium. This leads to a breach of this important brain protective barrier, allowing pathogens to leak into the cerebrospinal fluid and diffuse into the brain tissue [71,78]. So far, two barriers which protect the brain—the blood–brain barrier and the choroid plexus epithelial barrier—are broken. The ACE2 receptor spike protein interactions can also be increased by ischemic insults and inflammation in the brain. This in turn may lead to possible changes in blood pH as well as brain pH changes through hypoxemia [79].

In respect to the structural dynamics of the virus spike protein, we demonstrated with CAMM methods that pH-dependent effects of His residues and Zn^2+^ ions exist. Since incretin mimetics have been identified as potential therapeutics under special consideration of long-COVID symptoms we focused on two substances, Semaglutide and Pt2His. These substances have one or two His residues in their amino acid sequence. In this context, the impact of the body temperature (the second clinical parameter in addition to the blood pH value) raised our interest. It was not possible to detect effects on the structural dynamics by NMR methods within the small pH interval determined by clinical observations. However, it was possible in this small pH range of interest to examine effects of the blood and body temperature. It turned out that the fluctuations of the body temperature between 37 °C and 42 °C are associated with alterations in the structural dynamics of the four peptides in complex with the DPC micelles under study. It is not surprising to identify such temperature effects when studying linear peptides in a liquid environment. However, in our case, it was of importance in which way these linear peptides were able to interact with dodecylphosphocholine (DPC) micelles. We obtained different results for Semaglutide as well as for PtHis2 when ^31^P NMR experiments were performed at 42 °C and 37 °C (Figure 9). This documents the importance of temperature altering DPC-peptide interactions in the context of Pt2His and Semaglutide at these two body temperatures. A temperature of 42 °C represents the crucial point, especially when we correlate these findings with the ^31^P NMR data obtained for 2RUM and RUO in the presence of DPC micelles.

Our clinical observations together with CAMM methods indicate that pH and temperature effects are of interest when SARS-CoV-2 spike protein interactions with potential ligand structures are studied. Semaglutide with its primary His residue is an approved medication against diabetes and the incretin mimetic proto-type Pt2His is already a part of human clinical studies. In addition to having a positive effect in the field of diabetes therapy, these drugs are also applied in the field of neurological diseases, e.g., after brain traumata. This may infer that they may be of potential use in the case of SARS-CoV-2 infections where long-COVID symptoms may affect nervous tissues, especially where it involves the collagen network of the blood–brain barrier. These changes should be considered to be among the most dangerous threats of this viral infection. Since the mechanism by which the incretin mimetics under study are working is not yet known, some possibilities have to be considered. They could act as antiviral fusion peptides which destroy the platform for virus entry prepared by viral fusion peptides. They may accomplish this by interfering directly with the cell membrane or with surface-exposed gangliosides along with glycosylated GIP/GLP-1 receptors. Incretin mimetics can bind in a specific way to GIP/GLP-1 receptors on nervous cells. The receptor concentration is triggered by the glycosylation pattern of the incretin receptors [80]. Incretin mimetics have been employed in clinical applications to prevent or cure long-COVID symptoms in order to control the over-expression of cytokines [81].

Here, we see a strong link to the field of glycobiology where clinical questions related to SARS-CoV-2 infections and a potential impact on the nervous system may be addressed [82,83,84]. The therapeutic effects of lipid encapsulated inhibitory SARS-CoV-2 fusion peptides for injection or an oral administration are dependent on their membrane affinity. In this context, it was advantageous that incretin mimetics without (Tirzepatide) or one (Semaglutide) or two (Pt2His) pH-sensitive His residues exist (Figure 1). Incretin mimetics, in special cases, are used in addition to other therapies in TD2 as well as for treatments of mixed forms of TD2 and TD1 [85,86]. Furthermore, incretins are also applied during treatment of traumatic injuries [87,88]. Importantly, we note here that the presence of the incretin receptors (GIP and GLP-1 receptor) is regulated by N-glycosylation [80].

Answers to biomedical questions provided by tools applied in the field of nanomedicine such as quantum-chemical/DFT calculations on collagen–integrin interactions [89], as well as neuronal research related to sialic acids [20] and studies on the lectin-like function human lysozyme [7,8] may provide valuable hints for the direction of treatment options to curtail the SARS-CoV-2 pandemic. In a recent publication (retrospective study [90]), it was reported that incretin mimetics used in TD2 diabetes therapy had no effect on COVID-19--related symptoms in diabetic patients under incretin treatment infected by SARS-CoV-2. This is indeed not surprising because our study clearly shows that incretin mimetics with an altered amino acid sequence act in different ways with model membranes (being necessary to consider in relevant clinical studies). Based on this, it was necessary to analyze incretin mimetics which differ from Semaglutide, Liraglutide, Tirzepatide, Lixisenatide or other clinically applied incretin mimetics. We therefore focused on the comparison of the pH and temperature-dependent structural dynamics of Pt2His and Semaglutide in relation to glycosylated membrane models. Remarkably, the results obtained for Pt2His, shows significantly altered membrane interaction properties in comparison to Semaglutide. The differences between Pt2His and Semaglutide can in part be accounted for by structural dynamics in terms of temperature, solubility and membrane effects. The Pt2His sequence contains two pH-sensitive His residues at positions 1 and 3 that lead to different 1H and ^31^P NMR spectra in comparison to Semaglutide (Appendix A and Figure 9). In particular, Appendix A demonstrate that the solubility of these two incretin mimetics differs strongly in a watery or a lipid environment. This points to an altered membrane affinity making Pt2His a promising prototype in our search for inhibitory viral fusion peptides acting in opposition to the membrane perturbing SARS-CoV fusion peptides 2RUM and 2RUO. The altered membrane interactions could be a consequence of a higher affinity to glycosylated membranes of Pt2His due to the protonation state of the two His residues attracted by the negatively charged sialic acid residues.

Notably, NMR experiments were carried out under consideration of the two physiological key parameters, temperature and pH value. In addition, the presence of a glycosylated model membrane and sialylated silica nanoparticles as a simple virus model were also essential prerequisites to perform such experiments. The incretin mimetics Semaglutide and Pt2His as well as the two SARS-CoV fusion peptides 2RUM and 2RUO display a detectable model membrane affinity which is of interest when characterizing potential inhibitory viral fusion peptides. ^1^H (proton) NMR data and ^31^P (phosphorus) NMR data of DPC micelles mixed with ganglioside GM1 at various concentrations in the presence of Semaglutide, Pt2His, sialic acid-coated nanoparticles (Figure 9) as well as the SARS-CoV fusion peptides 2RUM (19-mer peptide) (Figure 10) and 2RUO (16mer peptide) (Figure 11 and Figure 12) were recorded with various NMR methods during our recent study. The Appendix A document the variations of the four peptides under study in respect to temperature, the solution and the presence of DPC and GM1 by 1H NMR methods.

In previous publications, we demonstrated that, in addition to a binding affinity to sialic acid moieties, a specific binding to sulfated carbohydrates is also important to establish stable ligand receptor complexes which are related to so-far non-answered bio-medical questions [91,92,93]. Furthermore, ab initio and DFT quantum-chemical calculations are necessary to obtain the nanomedical details of these affinities since they may depend on certain functional groups or conformational states [94,95] as well as on the stabilizing cations involved [89].

It has to be mentioned here that we are not restricted to DPC micelles. Triggering the fibrilization of model proteins such as human lysozyme and influencing the stability of phosphatidylserine emulsions are additional prerequisites [73,96,97] on our way to designing encapsulated inhibitory viral fusion peptides against MERS and SARS-CoV-2 [98,99,100].

Future cell tissue experiments with the before mentioned antiviral fusion peptides and incretin mimetics will be performed under various pH conditions (at approximately pH 6 on the lung surfaces and between pH 7,35 and 7,45 in the blood stream). Semaglutide with one and Pt2His with two His residues are here the most promising candidates because histidine are pH-sensitive residues under physiological conditions. Its imidazole ring changes at pH 6 from a positively charged ring system to an aromatic ring. Human serum albumin (HSA) is a suited shuttle system since the lipid tails of Semaglutide and Pt2His interacts in a specific way with the protein (1e7e.pdb) in the blood stream (Figure 3, Figure 4 and Figure 5).

Concerning the SARS-CoV fusion peptides, 2RUM and 2RUO, it is of interest to determine whether these two peptides react in a different way when a glycosylated DPC membrane model is analyzed with ^31^P and 1H NMR methods. The findings published here are a prerequisite for further MD supported NMR studies and a detailed analysis of how our data can be confirmed in human tissue probes [9,101,102,103,104,105,106].

Our observation strongly suggests that blood pH is an important parameter, which has to be considered when designing SARS-CoV-2 inhibitory fusion peptides. We have already noted that the pH sensitivity of the His residues at positions 1 and 3 of Pt2His are important in this context. This result provides a basis to further refine and optimize the sequence of the incretin mimetic prototype Pt2His as well as other SARS-CoV-2 inhibitory fusion peptides. It is a key step in combating this inflammatory disease, especially when focusing on the nervous system.

## 4. Materials and Methods

### 4.1. Chemicals

The model structures are two incretin mimetics (Semaglutide and Pt2His, a new incretin mimetic prototype) as derived from the corresponding amino acid sequence of the Gila monster.

These substances were analyzed together with the two SARS-CoV fusion peptides RUO and RUM [2] (16mer peptide, pdb code: 2RUO.pdb and 19-mer peptide, pdb code: 2RUM.pdb). Figure 3 illustrates the relevant sequences.

Semaglutide and Pt2His were provided by United Laboratories, China.

Two linear SARS-CoV-related fusion peptides were tested for comparison under similar conditions. The fusion peptides (equivalent to 2RUM.pdb; MWKTPTLKYFGGFNFSQIL-NH2 and 2RUO.pdb; GAALQIPFAMQMAYRF-NH2) were obtained from Nanjing Yuan Peptide Biotechnology Co., Ltd.: https://cn.kompass.com/c/nanjing-peptide-biotechnology-co-ltd/cn192988/, accessed on 20 December 2020.

The purity of all studied materials was 98% as verified by HPLC methods.

The GM1 ganglioside was purchased from Sigma Aldrich, 64293 Darmstadt, Germany.

The sizes of the nanoparticles vary in the range from 50 to 100 nm in diameter [107].

Synthesis of Fluorescent Silica Nanoparticles (FSNP) fluorescein 5-isothiocyanate (39 mg) was dissolved in anhydrous ethanol (16 mL) and (3-aminopropyl) trimethoxysilane (17 µL) was added while stirring. The mixture was stirred overnight at 42 °C to produce the precursor solution. For the synthesis of fluorescein isothiocyanate (FITC)-doped silica nanoparticles, 5.0 mL of the precursor solution was added to anhydrous ethanol (34 mL) while stirring followed by the addition of tetraethyl orthosilicate (1.7 mL) and aqueous ammonia (1.4 mL, 25% *v*/*v*). The mixture was stirred for 48 h to give the FSNP solution.

Synthesis of perfluorophenyl azide (PFPA)-functionalized Fluorescent Silica Nanoparticles (FSNP-PFPA) FSNP-PFPA was synthesized by adding a solution of silane-derivatized PFPA in toluene (7.0 mL, 10 mg/mL) into the FSNP solution and stirring at room temperature for 24 h. The product was purified by repeated washing and centrifugation in acetone (12,000 rpm of a HERMLE Labnet Z326 centrifuge, 30 min), and was re-suspended in 10 mL of acetone to give FSNP-PFPA.

Synthesis of sialic acid-conjugated FSNP (FSNP-Sia) To 1 mL of FSNP-PFPA in acetone, an aqueous solution of sialic acid (1.0 mL, 3.6 mg/mL) was added. The glass jar containing the mixture was covered with a 280 nm long-path optical filter and was subsequently irradiated with a 450 W medium-pressure mercury lamp for 40 min. The resulting FSNP-Sia was purified by repeated washing in Milli-Q water and centrifugation (12,000 rpm of a HERMLE Labnet Z326 centrifuge, 30 min). The final product was re-suspended in water.

### 4.2. NMR Spectroscopy

All NMR experiments were carried out with either a Bruker Avance II NMR spectrometer of 750 MHz or an Avance III NMR spectrometer of 600 MHz, both equipped with a regular probe-head. Lyophilized peptides were either dissolved in water, or in water with 200 mM DPC. Pure peptide solutions were characterized by regular 1D 1H and 2D TOCSY (60 ms mixing time), 2D NOESY (150 ms mixing time) and 2D 1H-13C HSQC experiments at 600 MHz, while peptide–DPC mixtures were measured at 750 MHz. Experiments were carried out at 37 °C and 42 °C. Interaction with GM1 was studied by addition of GM1 to final concentration of 0.5 mM. The 1D ^31^P NMR spectra were recorded at 700 MHz without 1H decoupling, using an acquisition time of 0.9 s, a recycle delay of 5 s, and 64 scans. All spectra were processed using Bruker Topspin software suite.

### 4.3. Computer-Aided Molecular Modeling (CAMM)

#### 4.3.1. ACE2-Receptor/Spike Protein and Collagen Complexes

The binding sites of the ACE2 protein were analyzed at first by the SiteMap [108,109] program of Schrödinger [110]. The ACE2-receptor/spike protein and collagen complexes were solvated in Schrödinger’s Maestro [111] with the water molecules added within a 1 nm buffer at neutral pH around the proteins. The resulting structures were then minimized and equilibrated for 5 ns. The final structures after equilibration were submitted for 50 ns NPT (pressure at 1.01325 bar) molecular dynamics (MD) simulations with the Desmond program [112] at 300 K. Molecular geometries resulting from simulations were saved at 10 ps intervals and were used for further analysis. Maestro was used for visualization of molecular complexes.

#### 4.3.2. De Novo Protein Structure Prediction and Homology Modeling of Pt2His and Semaglutide

For Tirzepatide, Semaglutide and Pt2His, no crystallographic structure data are available in the RCSB PDB database. Therefore, de novo molecular constructions with PEP-FOLD 3 [113] and Raptor [114] as well as homology modeling with SWISS-MODEL [115] were completed in order to build the 3D models. Consequently, molecular dynamics (MD) simulations were performed using YASARA program [116]. The hydrogen bond network [117] was optimized at first to increase the solute stability together with pKa prediction to fine-tune the protonation states of protein residues at the chosen pH of 7.4 [118] Na^+^ and Cl^−^ ions were added with a physiological concentration of 0.9% The structures thus created were minimized at first (steepest descent and simulated annealing). The structures were then solvated using the TIP3 water model. One hundred nanosecond MD simulations were performed using the AMBER14 force field [119] for the solute, GAFF2 [120] and AM1BCC [121] for ligands, special amino acids (hydroxyproline) and ions. The cutoff was set to 8 Å for Van der Waals forces (the default used by AMBER [122], no cutoff was applied to electrostatic forces (using the Particle Mesh Ewald algorithm [123]. The equations of motion were integrated with a multiple time step of 1.25 fs for bonded interactions and 2.5 fs for non-bonded interactions at a temperature of 310 K, a density of 0.993249 g/mL and a pressure of 1 atm (NPT ensemble) in accordance with previous algorithms [124]. The first 10 ns (equilibration) of the simulation time after inspection of the solute RMSD was excluded from further analysis.

In addition, RMSD (Appendix A) and RMSF (Appendix A) values for the individual residues of Tirzepatide, Semaglutide and Pt2His were calculated.

### 4.4. Statistical Analysis

The statistical analysis was performed using the SPSS 25.0 software.

## 5. Conclusions

We would also like to emphasize that the mechanism described in our study offers solutions for virus-induced pathology in general. We tested potential inhibitory SARS-CoV-2 fusion peptides together with SARS-CoV fusion peptides in the presence of a glycosylated model membrane and sialic acid-coated nanoparticles as a simple virus model. The sialic acid affinity provides key information on how inhibitory peptide sequences can be further optimized given the fact that this contact structure is present on the SARS-CoV-2 spike protein as well as on the host cell membrane.

This opens a new route to a potent weapon against SARS-CoV-2 variants that can also be directed versus future as yet unknown viral species.

In summary, vaccination is a strong tool to combat viral infections. However, it is also important to develop alternative strategies that will enable clinicians to fight quickly against the evolving severe threats imposed by mutations within the pathogen. A promising strategy for this is the application of inhibitory viral fusion peptides which destroy the platform on the host cell membrane created by the virus itself in order to foster its entry. A first attempt towards this step may involve incretin mimetics, which have already been clinically tested for a number of medical applications (e.g., approaches in therapies against diabetes and brain traumata). The lipid side-chains of incretin mimetics are known to bind to serum albumin. Once bound, they are carried to various tissues and cell types by the circulatory system [125], where they interact with GIP and GLP-1 receptors on the cell. This means that aside from targeting the islet cells of the pancreas, incretins target cells in other tissues throughout the body.

It is still not yet known whether the interactions of certain incretin mimetics which contact various cell membrane structures or specific receptors located on the membrane are mediated by sialic acid residues or other carbohydrate moieties. Sialic acid-related contact structures are present on the SARS-CoV-2 spike protein as well as on the host cell membrane. In particular, the protonation state of histidine residues plays a crucial role in this context. In respect to the fate of patients in intense care units, we focused on incretin mimetics having one (Semaglutide) or two (Pt2His) histidine residues. The substances under study are temperature and pH sensitive in the physiological range. Our results indicate that the arsenal of methods provided by structural biology including molecular modeling tools will be invaluable in finding optimized inhibitory viral fusion peptides to combat pandemic viral diseases.

The strategy discussed here can be considered as an important step in developing pan-incretins against SARS-CoV-2 mutations, long-COVID symptoms and other neurological diseases such as multiple sclerosis [84,126,127,128,129,130]. The application of linear peptides is discussed as a potential therapy that could block virus entry by interfering with the host cell membrane and/or with the glycosylated virus spike proteins. The advantage of incretin mimetics in this context is their approved use in clinical applications mainly in correlation with diabetes. There are a number of incretin mimetics available on the global market and the individual amino acid sequences are responsible for altered binding properties. Therefore, we have compared a well-established incretin mimetic (Semaglutide) and a new incretin mimetic prototype in which the amino acid sequence has been modified in accordance to clinical and biophysical observations.

Concerning myalgic encephalomyelitis/chronic fatigue syndrome, there was no selection here. We analyzed a presumably representative sample. All COVID-19 patients in the hospital requiring intensive care were included at the time of the examination. An accumulation of pre-existing myalgic encephalomyelitis/chronic fatigue syndrome in the sample is therefore almost impossible. The results obtained from the patients in the intense care units allowed us to initiate studies related to patients who developed long-COVID conditions. This is the reason why we focus now on the analysis of incretin mimetics which are approved for diabetes therapies and treatments in the case of brain injuries. These results will be published later on.

The symptoms of long COVID are known to be related to various inflammatory diseases. Indeed, in another trial (unpublished yet), we analyzed inflammatory markers and the outcome was comparable to our current analysis of pH values. Concordant with the pH values and the corresponding analysis, we detected an impact of inflammatory markers and other parameters associated with an adverse outcome as well, i.e., IgM, CRP, leukocyte values, PCT, IL-6, LDH, D-Dimers, and Ferritin.

## Data Availability

The data presented in this study are available on request from the corresponding authors.

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
