# Peer review of "Blood pH Analysis in Combination with Molecular Medical Tools in Relation to COVID-19 Symptomsâ€"

_biomedicines, 2023, doi:10.3390/biomedicines11051421_

Round 1
Reviewer 1 Report
Siebert et al studied the blood pH level of COVID-19 patients and showed that critically ill and fatal patients have lower blood pH level than survived patients. They attributed it to the His residue interaction of ACE2 receptor that would modify the binding of SARS-CoV2 spike protein. They characterized incretin-mimetics with SARS-CoV2 peptides interactions with NMR and CAMD and provided a basis for small molecule drug development against SARS-CoV2. Here are the following comments to be addressed before publication.
1. In the abstract, they should clearly mention that blood pH is lower in fatal COVID-19 patients because it is one of the major result.
2. The last part of the introduction should be concise only with relevant points to the goal of the paper. It appears descriptive.
3. Line 152-153 should be in methods, but not in introduction.
4. Line 185, what is the implication of ACE2-SPIKe binding alterations with long COVID symptoms? If any relation is there, they should mention it.. Authors also used long COVID symptoms frequently in other places (line 374). I think it would be COVID symptoms
5. In Figure 1, it seems 19 patients (10 fatal + 9 survived) but authors wrote for 29 patients in the text (line 211, 217). What are the actual numbers? Texts should be corrected. Again there is no asterix (*) for p value in the figure. These should be mentioned or corrected.
6. Same 19 patients in figure 2. But legend indicates 29 patients.
7. Line 226-234, this paragraph should moved to NMR experiment section as a rationale.
8. In Line 273, it would be HSA (Human Serum Albumin) instead of HAS.
9. MD as molecular dynamics is abbreviated much later in the text in line 367 but referred many times earlier. It should be mentioned when first used.
10. Line 376-389 should be moved to rationally of NMR experiments.
11. In line 564-571 and in earlier other places, author always mentioned the implication of pH changes and SARS-Cov2 infection for BBB. Is it due to contact of blood and cell communication? Because in all SARS-ACE2 interaction inother cells also could be impacted by blood pH as SARS-CoV2 travels through blood and infect cells where ACE2 expresses. The y should mention or discuss in general. Only for collagen, they should mention BBB.
12. Discussion is too long. They should only discuss the implication of their results and compress it. Overall writing is too descriptive in whole manuscript. Authors should present their results concisely and discuss relevant points. Moderate English editing is required. Large sentences should be breakdown for easy understand, such as, Line 106-109.
Author Response
- In the abstract, they should clearly mention that blood pH is lower in fatal COVID-19 patients because it is one of the major result.
This fact is now explicitly mentioned at the end of the abstract.
- The last part of the introduction should be concise only with relevant points to the goal of the paper. It appears descriptive.
The introduction has been changed appropriately as advised.
- Line 152-153 should be in methods, but not in introduction.
This text has been moved to the methods section.
- Line 185, what is the implication of ACE2-SPIKe binding alterations with long COVID symptoms? If any relation is there, they should mention it. Authors also used long COVID symptoms frequently in other places (line 374). I think it would be COVID symptoms
More patients need to be studied to clarify this, and more accurate results will come in a subsequent publication.
- In Figure 1, it seems 19 patients (10 fatal + 9 survived) but authors wrote for 29 patients in the text (line 211, 217). What are the actual numbers? Texts should be corrected. Again there is no asterix (*) for p value in the figure. These should be mentioned or corrected.
Figure 1 shows the time dependence of the average pH of two groups of Covid patients, i.e. 12 fatal cases and 13 survivors. The total number of patient was 25, not 29. The number is now corrected throughout the text. The numbers 10 and 9 correspond to days when blood samples were collected and pH measured.
- Same 19 patients in figure 2. But legend indicates 29 patients.
The dates shown in Figs 1-3 are average values for all patients based on the number of days each patient was in the ICU. The x-axis shows days on all these figures
- Line 226-234, this paragraph should move to NMR experiment section as a rationale.
The paragraph was moved to the NMR section as requested.
- In Line 273, it would be HSA (Human Serum Albumin) instead of HAS.
This was misspelled in the original text and has been now been corrected.
- MD as molecular dynamics is abbreviated much later in the text in line 367 but referred many times earlier. It should be mentioned when first used.
We added the following sentence to the end of the Introduction. “The methodology used is consistently described in part 4 of the presented article.” Accordingly, the MD abbreviation for Molecular dynamics is mentioned in part 4 of the article.
- Line 376-389 should be moved to rationally of NMR experiments.
The advice was accepted.
- In line 564-571 and in earlier other places, author always mentioned the implication of pH changes and SARS-Cov2 infection for BBB. Is it due to contact of blood and cell communication? Because in all SARS-ACE2 interaction in other cells also could be impacted by blood pH as SARS-CoV2 travels through blood and infect cells where ACE2 expresses. The y should mention or discuss in general. Only for collagen, they should mention BBB.
Thank you for this constructive note. We will look at this in more detail in future studies.
- Discussion is too long. They should only discuss the implication of their results and compress it. Overall writing is too descriptive in whole manuscript. Authors should present their results concisely and discuss relevant points. Moderate English editing is required. Large sentences should be breakdown for easy understand, such as, Line 106-109.
Thanks for this comment and all other comments mentioned above. The Discussion was shortened and style was updated by our coauthor John W. Hudson, who is a Canadian citizen with English family background and education.
Reviewer 2 Report
ID: biomedicines-2294535: review
Blood pH Analysis in Combination with Molecular Medical Tools in Relation to Long Covid Symptoms. by Siebert et al.
To the Authors:
General comments:
The authors investigated the correlation of clinical biomedical data derived from patients in intensive care units with structural biology and biophysical data from NMR and/or CAMM (Computer Aided Molecular Modeling). They evaluated the suitability of incretin-mimetics with one or two pH-sensitive amino acid residues as potential drugs to prevent or cure long COVID symptoms. It was considered that the topic was interesting, and the results included novelty; however, several points should be addressed to improve the manuscript.
Specific comments:
1. The authors should show how the present patients have developed into long COVID condition, since long COVID occurs 2-3 months after the acute infection. The patients treated in ICU do not always match the conditions of long COVID.
2. The symptoms of long COVID are known to be related to various inflammatory diseases including myalgic encephalomyelitis/chronic fatigue syndrome. Please describe the clinical histories of the present patients.
3. Did the authors evaluate the relationship between pH and serum levels of various inflammatory markers, electrolytes, blood glucose and other metabolic factors?
4. Please discuss the possibility that antiviral agents used for the patients in ICU have affected the molecular and biophysical data. How about the relationship between the biophysical data and the levels of serum SARS antibody? Differences between the viral variants and the effects of COVID vaccination should also be discussed.
Author Response
- The authors should show how the present patients have developed into long COVID condition, since long COVID occurs 2-3 months after the acute infection. The patients treated in ICU do not always match the conditions of long COVID.
Since 25 patients represent a rather small group, we are now analyzing the fate of many more people suffering from long COVID symptoms in relation to a treatment with incretin-mimetics. The results will be published in the framework of a follow-up project. In this follow-up project we will also analyze the relationship between pH and serum levels of various inflammatory markers, electrolytes, blood glucose and other metabolic factors. The therapeutic approaches described in this article document the situation at the start of the pandemic event. Antiviral agents and vaccination were not approved at that time. The goal here was to collect clinical and biophysical data (NMR and CAMM) and employ this information along with the use of SARS CoV fusion peptides in order to find new strategies to combat the development of COVID cases that may become critical. The data gained here will be applied in our studies on long-COVID symptoms.
There was no follow up on these patients so we cannot answer this question. Furthermore, follow-ups would have vastly increased the complexity of the study and would have required new ethics approval. This would have substantially delayed our study. Above all the short-term effect of the pH value on the outcome, i.e. survival was the main question and not the development of Long Covid.
- The symptoms of long COVID are known to be related to various inflammatory diseases including myalgic encephalomyelitis/chronic fatigue syndrome. Please describe the clinical histories of the present patients.
Concerning a myalgic encephalomyelitis/chronic fatigue syndrome, there was no selection here. We analyzed a presumably representative sample. All COVID patients in the hospital requiring intensive care were included at the time of the examination. An accumulation of pre-existing myalgic encephalomyelitis/ chronic fatigue syndrome in the group of patients under study is unlikely. It was not possible within the scope of the study to analyze this interesting point but will be considered in a follow up-project.
The ethics approval allowed the evaluation of pH, temperature and outcome. Therefore, additional data was not recorded. There was no selection here in the choice of patients. Presumably this was a representative sample, since all COVID patients in the hospital requiring intensive care were included at the time of the examination. An accumulation of pre-existing myalgic encephalomyelitis in the sample was therefore almost impossible.
- Did the authors evaluate the relationship between pH and serum levels of various inflammatory markers, electrolytes, blood glucose and other metabolic factors?
In another trial (unpublished yet) we analyzed inflammatory markers and outcome comparable to our current analysis of pH. Concordant with pH values and outcome, we detected that inflammatory markers and other parameters are associated with adverse outcome as well, i. e. IgM, CRP, Leukocyte, PCT, IL-6, LDH, D-Dimers, Ferritin.
- Please discuss the possibility that antiviral agents used for the patients in ICU have affected the molecular and biophysical data. How about the relationship between the biophysical data and the levels of serum SARS antibody? Differences between the viral variants and the effects of COVID vaccination should also be discussed.
Molecular and biophysical data was collected independently of the pH values of the patients. Antiviral agents were not routinely used in the early phase. The determination of the SARS antibodies was not possible at this time (early phase) and was therefore not carried out. Only the wild type was tested, no variants and no vaccination. In this respect, this can only be discussed theoretically.
Round 2
Reviewer 1 Report
The manuscript is now much improved. But I still have some minor concerns that should be corrected.
1. In Figure 1 and 2 legends, authors should write that each point in the graph represents the average value of measurement of pH at that particular day.
2. Long covid should be changed to COVID-19 (syndrome/symptoms). Because this effect of pH or ACE2 binding may occur in general COVID-19 and may be with long COVID. It is not understandable why authors are sticking/relating particularly with long covid for the effect.
Author Response
- In Figure 1 and 2 legends, authors should write that each point in the graph represents the average value of measurement of pH at that particular day.
The comment was incorporated
- Long covid should be changed to COVID-19 (syndrome/symptoms). Because this effect of pH or ACE2 binding may occur in general COVID-19 and may be with long COVID. It is not understandable why authors are sticking/relating particularly with long covid for the effect.
The comment was incorporated
Reviewer 2 Report
ID: biomedicines-2294535: 2nd review
Blood pH Analysis in Combination with Molecular Medical Tools in Relation to Long Covid Symptoms. by Siebert et al.
General comments:
The authors revised many of issues based on the referee’s suggestion, although there still be some points to be addressed to improve the manuscript.
Author Response
- The authors should show how the present patients have developed into long COVID condition, since long COVID occurs 2-3 months after the acute infection. The patients treated in ICU do not always match the conditions of long COVID.
Yes, this is correct. As a result of the data discussed here we have initiated a clinical study with diabetic long-Covid patients which are treated with incretin-mimetics.
- The symptoms of long COVID are known to be related to various inflammatory diseases including myalgic encephalomyelitis/chronic fatigue syndrome. Please describe the clinical histories of the present patients.
We have already addressed this important suggestion in the text. These are important hints which will be considered in the follow-up study mentioned above.
- Did the authors evaluate the relationship between pH and serum levels of various inflammatory markers, electrolytes, blood glucose and other metabolic factors?
Will be carried out in the follow-up study mentioned above
- Please discuss the possibility that antiviral agents used for the patients in ICU have affected the molecular and biophysical data. How about the relationship between the biophysical data and the levels of serum SARS antibody? Differences between the viral variants and the effects of COVID vaccination should also be discussed.
This is now discussed in the text. We have to emphasize that these clinical data are obtained at the beginning of the pandemic event in the year 2020.